# The Effects of Investment in Major Construction Projects on Regional Economic Growth Quality: A Difference-In-Differences Analysis Based on PPP Policy

**Fang Wang** [1,2], **Ming Yao** [1], **Xianhua Huang** [2], **Hao Guo** [3], **Penghui Zheng** [3] **and Hongwei Yu** [4,*]

[1] Ningxia Coal Industry of China Energy Group, Yinchuan 751400, China; wanfy015@126.com (F.W.); 15047390@chenergy.com.cn (M.Y.)
[2] China Energy Inverstment Group, Beijing 100011, China; xianhua.huang@chnenergy.com.cn
[3] Economics and Management School, Wuhan University, Wuhan 430072, China; m13625680301@163.com (H.G.); 15927314417@163.com (P.Z.)
[4] Institute of Quality Development Strategy, Wuhan University, Wuhan 430072, China
[*] Correspondence: yuhongwei928@163.com

**Abstract:** The investment in Major Construction Projects (MCPs) has a counter-cyclical impact on quantitative GDP increases during the recession period. However, its impact on the quality of economic growth is still unknown. Based on the data of prefecture-level cities in China from 2008 to 2017, we construct an economic Quality Growth Index (QGI) including sustainable development factors and take the PPP (Public–Private Partnership projects) policy as a quasi-natural experiment to design a Difference-In-Differences (DID) strategy for the first time to estimate the effects of the MCPs investment on the sustainable development of regional economies. We find that the MCPs investment can significantly improve the quality of regional economic growth. The MCPs investment can improve the quality of regional economic growth by enhancing innovation and entrepreneurship at the regional levels. Our findings may provide empirical evidence to support the policy of increasing investment into infrastructure constructions to promote sustainable development in the current economic recession under the COVID-19 pandemic.

**Keywords:** major construction project; economic growth quality; PPP policy; regional entrepreneurship; Difference-In-Differences

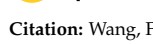



## 1. Introduction

Nowadays, the global economy is in recession under the impact of the COVID-19 pandemic, and many governments are making efforts to seek ways to drive a fast economic recovery. Since the investment in Major Construction Projects (MCPs) plays an important role in regional economic development [1], especially during the recession period, governments worldwide tend to adopt the investment of MCPS as a counter-cyclical adjustment economic policy tool for a rapid GDP increase in a short term [2,3].

However, governments should consider the overall economic growth quality other than the short-term quantity increase in GDP when they come to the decision of MCPs investments, which is more beneficial to the sustainable development of economies. Much existing literature has explored the effects of MCPs on regional economic growth from different perspectives, such as the transportation infrastructures [4–7], the communication facilities [8,9], the power infrastructures [10–12], etc., and most of the results have proved that the MCPs investment can help the regional economy to achieve an immediately positive increase in GDP by expanding regional employment [13], reducing the economic transaction cost [14] and improving the efficiency of goods circulations [15]. Some other literature also indicates that a large number of infrastructure constructions may aggravate the debt of governments [16], reduce the financial support to manufacturing sectors [17], and squeeze out other producing resources [18], which may be not so conducive to the

long-term economic development. It is not wise for the governments to focus on the short-term counter-cyclical adjustment effects of MCPs investment but ignore its impact on the long-term sustainable development when encountering the pressure of the COVID-19 pandemic on regional economic recovery.

This paper will investigate the overall impacts of MCPs investment on the regional economic sustainable development and provide useful decision-making references for the governments to stimulate the regional economy in the current COVID-19 pandemic situation. Some studies conclude the positive effects of MCPs investment on regional long-term economic growth include improving the ecological environment [19], optimizing the industrial structure [20], and promoting the innovation capacity [21], as well as the negative outcomes such as the misallocation of financial resources [22], increasing the financial risk [23,24]. Those exiting conclusions only reveal the impacts on some specific aspects of long-term economic growth, which is not an "overall" assessment and not conceivable enough to support government decisions that are usually based on holistic considerations. Therefore, it is necessary to construct a comprehensive index to measure the sustainable development of the regional economy to further explore the impact of MCPs investment.

We will construct an economic Quality Growth Index (QGI) to figure out this "overall" effect. The difference between the quality of economic growth and traditional economic growth mainly lies in the fact that the quality of economic growth aims at sustainability rather than simply increasing the quantitative GDP growth rate [25]. In terms of the measurement of the high-quality development of regional economy, some literature documents that economic growth efficiency is the core of improving economic growth [2]. Therefore, total factor productivity is used as a proxy variable of the quality of economic growth [16,19]. However, the evaluation of the quality of economic growth should include multiple aspects, such as economic growth rate, social welfare, environmental protection, etc. [24]. Based on the study of Mlachila et al. [26] on the establishment of economic QGI, including both the quantity of economic growth and the dimension of social attributes, we have further incorporated environmental factors into the index system to measure the sustainable development level of the regional economy.

In order to accurately capture the effect of MCPs investment on the overall quality of regional economic growth, we need to find an adequate strategy to deal with some possible endogenous problems between MCPs investment and regional economic growth, such as the possibility that regions with better quality of economic growth tend to increase the MCPs investment, which will cause our estimating results biased. In this paper, we take the Public–Private Partnership (PPP) project policy implemented in China around 2014 as a quasi-nature experiment to design a Difference-In-Differences (DID) strategy to address the possible endogenous problems. The PPP projects in China are usually those infrastructure construction projects with huge investments, which are generally difficult to be accomplished by the single public sector or the private company and hence need the cooperation between them [27]. In this paper, we regard the PPP projects as the specific MCPs and will discuss more details about this in the following Section 2.

We use the prefecture-city level panel data from 2008 to 2017 in China for estimation. The results show that the investment of PPP can significantly improve the regional economic QGI. After the common trend test and the Propensity Score Matching–Difference-in-Differences (PSM–DID) estimation, the results stand robust. We further test the possible mechanism and find that the MCPs investment can enhance the regional innovation entrepreneurship so as to improve the quality of regional economic growth.

There are three marginal contributions of our paper. First, we explore the effects of MCPs investment on the overall economic development quality. Previous studies mainly focused on the short-term influence on the quantitative GDP increase [28,29] or the long-term impact on some specific aspects of economic growth, such as the misallocation of financial resources [22], increasing the financial risk [30]. In this paper, we focus on the "overall" effects on sustainable economic development by constructing the economic QGI to measure

the economic development quality, which is more helpful in comprehensively assessing the effects of MCPs investment on economic growth. Second, to our best knowledge, it is the first time to take the PPP policy as a quasi-natural experiment to carry out a DID strategy for the identification of MCPs investment in this paper. Existing literature tends to adopt the total investment to construct the year-and-city double fixed effects models to identify the effects of MCPs on economic growth [31], whose results are less conceivable due to their neglect of the possible endogenous problems. Third, we reveal a possible mechanism of MCPs investment promoting the quality of long-term economic growth, which is hardly involved in previous studies. In this paper, we find that the PPP investment can enhance regional innovation entrepreneurship so as to be conducive to the regional economic QGI. Our findings may provide empirical evidence support for governments to make the investment policy in the infrastructure constructions to promote the regional sustainable development, especially in the current economic recession after the COVID-19 pandemic.

The rest of the paper is organized as follows: Section 2 introduces the policy background of PPP projects and the theoretical basis; Section 3 is the research design; Section 4 is the conclusion and analysis; Section 5 is the verification of the mechanism; Section 6 is the conclusion.

## 2. Policy Background and Literature Review

### 2.1. PPP Policy Background

In order to meet the growing demand for public services and infrastructure, the PPP mode has been widely applied all over the world [32]. PPP mode refers to the cooperation between government departments and private companies, especially in the field of public infrastructure construction [33]. PPP mode has obvious advantages in dealing with insufficient government budgets and improving technology and management skills in MCPs [23,34]. It can not only provide new channels to cope with funding shortages in MCPs but also help to expand the investment of private capital and invigorate the construction market [29].

Although PPP projects are common in many developed countries in the world [35], they started late in China. We have manually collected and compiled the PPP policies and documents from China's central government from 2000 to 2018. As shown in Table 1, we can find that the Chinese government began to pay attention to and introduced a series of policies to vigorously promote PPP projects in the year of 2014. In addition, the Chinese government has built an information center for managing the PPP projects, named China Public Private Partnerships Center, and requires all PPP projects to be registered on this information center and disclose some information. We can freely download our research data on PPP projects in all prefecture-level cities from the website of the information center.

This paper adopts the PPP projects policy since 2014 to identify the MCPs investment mainly based on the following reasons. First, the characteristics of PPP projects are highly similar to those of the MCPs. PPP projects are usually public infrastructure construction projects that cover 19 sub-industries in various fields, including transportation, telecommunication, urban infrastructure, water conservancy and hydropower, environmental remediation, and so on [36]. They play a very basic and important role in supporting economic and social development. What is more, the investment in those construction projects tends to be huge, and it is difficult for the public sector to afford it alone.

Second, since the Chinese government began to vigorously implement the PPP projects in 2014, there were few PPP project investments in cities before 2014. If so, the PPP policy that began in 2014 can be regarded as a quasi-nature experiment for accurate identification. We count the number of all PPP projects by year from 2002 to 2018, as shown in Figure 1. We can easily find that before 2014 the number of PPP projects was very small, but in 2014 there was a sudden and sharp increase which is due to those policies issued in 2014, and then the number gradually decreased after 2015 but still maintains a certain amount. The sudden increase trend in 2014 allows us to take the cities implementing PPP projects investments as the experimental group and the cities without PPP investments as the control groups to

construct a DID strategy for estimation. Even though, after 2015, the trend goes downward, as long as there is a certain amount of PPP investment cities, it will not disable our divisions of experimental group and control group. In addition, we will conduct the parallel trends test in Section 4 to verify the effectiveness of our designed DID strategy.

**Table 1.** Chinese PPP Policies from 2014 to 2017.

| Time | Relevant Policies and Documents | Content Summary |
|---|---|---|
| 2014.9 | Notice on Issues Related to The Promotion and Application of The Government-Private Capital Cooperation Model | Accelerate the transformation of government functions, form a good system as soon as possible to promote the development of PPP mode |
| 2014.10 | Opinions on Strengthening the Management of Local Government Debt | Strengthen management of local government debt, and establish a unified oversight and management mechanism for borrowing, using, and repaying local government debt |
| 2014.11 | Guidelines on Innovating Investment and Financing Mechanisms in Key Areas to Encourage Nongovernmental Investment | Implement a unified market access system, make innovations in investment operation mechanisms and financing methods, and improve pricing mechanisms |
| 2014.12 | Guidelines on Cooperation between The Government and Private Capital | Encourage and guide social capital to enter the field of public construction, promote the adjustment of industrial structure |
| 2015.3 | Notice on Promoting Cooperation between The Government and Private Capital in Support of Development Finance | Establish a diversified and sustainable PPP fund guarantee mechanism through the flexible use of various financial instruments |
| 2015.4 | Measures for the Administration of Franchising of Infrastructure And Public Utilities | Encourage and guide social capital to participate in public service and infrastructure construction, and improve the quality of public services |
| 2016.1 | Law of the People's Republic of China on Cooperation between Government and Social Capital (Draft for public comments) | It is the first law in the field of PPP mode to be released to the public for advice, strengthening coordination and cooperation among various departments |
| 2016.5 | Notice on Further Enhancing Public–Private Partnership (PPP) Work | Raise financing efficiency and improve investment return mechanism |
| 2017.11 | Notice on Standardizing the Management of the Project database of the Comprehensive Information Platform for Public–Private Partnership (PPP) | Strictly standardize the operation of PPP projects and curb the risks increased in hidden debt |

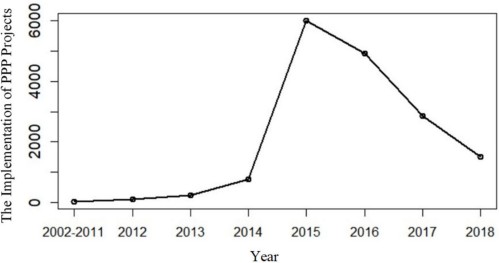

**Figure 1.** The Annual trend of PPP project number.

*2.2. Literature Review*

2.2.1. Impacts of MCPs Investment on Regional Economy

Most literature believes that the investment of MCPs plays an effective role in the regional economy in various aspects, such as driving regional GDP increases [4], offering new employment opportunities [12], adjusting industrial structure [5], stimulating technological innovation [37], and promoting human capital accumulation [38]. As a result, it becomes an important factor in promoting regional and national economic growth.

The existing studies many explore the impacts of MCPs investment on the regional economy from both short-term and long-term aspects. In the short term, it is well-known that the MCPs investment can play a counter-cyclical adjusting role in regional economic growth [13,39]. When the economy is in recession, a large amount of investment in MCPs may help to maintain a positive increase in regional GDP [37,40], as well as attract more employment workforce [32], thus alleviating the negative impact of the recession cycle on the economy.

In the long term, the MCPs may have a profound impact on regional economic development. Dong Chen [3], Liu Hongjuan et al. [41], and Wang Junli et al. [22] concluded that there is a clear connection between investment in infrastructure construction and industrial structure adjustment. The investment in infrastructure construction of transportation, post and telecommunications, and other aspects could promote the long-term optimization and upgrading of industrial structure and have a positive impact on future regional economic development [42]. Some other literature highlights the positive impact of MCPs on regional innovation. For instance, the construction and improvement of transportation or telecommunications infrastructure will reduce the cost of face-to-face trade among different regions, which may help the innovators to obtain tacit knowledge and promote their R&D cooperation [31]. Similarly, information technology infrastructure is also an important factor in promoting technological innovation because it helps to enhance corporate performance and improve human capital [43].

However, some literature also indicates that MCPs may negatively influence regional economic development. Since the MCPs investment mainly comes from the government fiscal or bank financial funds, it will squeeze out the capital input to other industries such as the manufacturing sectors, which may lead those industries to become more difficult in the recession [22]. Large investments in the short term can also exacerbate government fiscal deficits and bank financial risks [32]. Those producing resource misallocation and capital risks caused by short-term MCPs investment will also bring many adverse effects on the long-term regional economic growth, such as the deeper recession of the manufacturing sectors [29], the aggravation of economic bubbles [34], and so on.

Although the existing literature has investigated the positive and negative impacts of the MCPs investment on the regional economy both in the short-term and long-term, there is not enough evidence involving an "overall" impact on the sustainable development of the regional economy. The short-term impacts mainly focus on the quantitative increase in regional GDP, while the long-term effects are on some specific aspects of regional economic growth, including employment [12], industrial structure [43], technological innovation [25], financial risks [38] and resource misallocation [44], which are not conceivable to support the MCPs investment, especially in the recession. It is necessary to figure out the comprehensive effects of the MCP's investment on the sustainability of regional economic development.

2.2.2. Factors Promoting the Regional Economic Growth Quality

In this paper, we will use the regional economic growth quality to explore the "overall" effect of the MCP's investment on economic development. The difference between the quality of economic growth and traditional economic growth mainly lies in the fact that the quality of economic growth aims at long-term sustainability rather than simply increasing the quantitative GDP growth rate [25].

In recent years, scholars have intensively explored the ways of improving the quality of economic growth. The optimization and upgrading of the industrial structure are

regarded as the main drivers of China's rapid economic growth [45]. Human capital also has a positive impact on improving the quality of economic growth [46]. Hao Ying et al. [13] believed that corporate investment activities and the quality of economic growth are connected closely to each other. Shi Zili [47] indicated that the innovation system could effectively promote the quality of economic growth in the short term.

Since the investment of MCPs can promote the regional innovation capacity, we focus on the strand of literature supporting that innovation entrepreneurship promotes the quality of economic growth. Innovation entrepreneurship often represents the whole vitality of innovation in a region and plays an important role in improving the level of regional innovation [48]. Some scholars even believe that it is the core element of high-quality economic development [38,44]. The continuous improvement of entrepreneurship can create more resources for regional economic growth [49], including investment in producing factors to offer products and services that meet market demand, creating new employment opportunities [36], revitalizing industrial clusters and enhancing R&D activities [50], etc. Regional innovation entrepreneurship is also regarded as an important factor in driving sustainable growth, and it can improve the quality of economic growth mainly through technological innovation, industrial innovation, and organizational innovation [17]. Technological innovation can help enterprises develop and expand in the direction of higher factor input utilization and output efficiency, ultimately improving the production efficiency of the entire society and the quality of economic growth [51]. Entrepreneurial innovation is also beneficial to realize the upgrading of industrial technology, guiding the transfer of industrial structure to an advanced level and making the enterprises in the market more competitive [52]. Organizational innovation is helpful to improve resource utilization efficiency at the social level, which may improve the ecological environment while economic growth and achieve long-term sustainable development [48].

Combining the impact of MCPs investment on regional innovation and the effect of regional innovation entrepreneurship on the economic growth quality, we speculate that the investment of MCPs will promote the quality of regional economic growth by improving the regional entrepreneurial innovation. We will empirically examine this possible mechanism in Section 4.

### 3. Methodology

*3.1. Model*

Existing studies tend to use the panel fixed-effects model to estimate the impacts of investments on regional economic growth [50,51], which ignores the possible endogenous problems between investments and economic growth. The MCPs investment may lead to the regional economic growth, but on the contrary, the regional economic growth may possibly improve the investments in MCPs. The panel fixed-effects methods cannot distinguish the reverse causality and thus cause the estimated results bias.

In order to deal with the possible endogenous problems and obtain an unbiased estimation, we take the PPP projects policies since 2014 as a quasi-natural experiment to construct a time-varying DID strategy for identification. Our DID strategy divides the sample cities into experimental group and control group only according to the PPP policies since 2014 and regrades other characteristics remaining a continuous trend, which allows us to gain the unbiased effects of MCPs investment on economic growth quality by comparing the changes of the regional economic growth quality between the experimental and control groups before and after the policies.

We set our estimating model as follows:

$$QGI_{it} = \alpha_0 + \alpha_1 PPP_i * post_t + \alpha_2 X_{it} + \delta_t + \mu_i + \varepsilon_{it} \tag{1}$$

where, $QGI_{it}$ represents the quality index of regional economic growth of city $i$ in year $t$. $PPP_i$ is the dummy variable measuring the experimental group and the control group. It equals 1 if city $i$ has PPP projects investments; otherwise, it equals 0. $post_t$ is the dummy variable for policy implementation year. It equals 1 when it begins to invest in the major

PPP projects in year $t$, otherwise it is 0. $PPP_i * post_t$ is the DID interaction item which is measuring the effects of PPP projects shocks. $X_{it}$ represents a series of city characteristic variables that may relate to the quality of economic growth. $\delta_t$ and $\mu_i$ represent time fixed effect and city fixed effect, respectively. $\varepsilon_{it}$ is the error item.

### 3.2. Variables and Data

### 3.2.1. Explained Variables

Our purpose is to figure out the "overall" effects of MCPs investment on regional economic sustainable development. We construct a comprehensive index named the Quality of Growth Index (QGI) based on the high-quality economic growth framework of Mlachila et al. [26]. Mlachila et al. [26] agreed that high-quality growth was not only a high-speed growth but more meaningful to sustainable and socially friendly growth. They established a comprehensive set of indicators covering both the GDP nature and social attributes of economic growth. Compared with the Human Development Index (HDI) developed by the United Nations [20], their indicators are more helpful in assessing the quality of various growth stages within a country or a region [26].

Following the logic of Mlachila et al. [26], our QGI also includes the intrinsic GDP and social dimensions. Among them, the intrinsic nature of growth includes the four metrics of economic growth: intensity, stability, the diversity of sources, and the degree of extroversion. The speed of economic growth is measured by the annual increase in GDP per capita. The stability of economic growth is measured by the inverse of the coefficient of variation of the growth rate of GDP per capita, and the coefficient of variation is the ratio of the standard deviation to the mean. The larger the coefficient of variation, the worse the stability of economic growth. The diversity of sources of economic growth represents the degree of growth generated by various sources. This paper uses the *Theil index* to measure the diversity of economic growth in cities. The degree of extroversion of economic growth refers to the share of net external demand, which is measured by the ratio of the total import and export volume to the GDP.

Another dimension of QGI is the social indicator of economic growth. Mlachila et al. [26] pointed out that the measurement of the quality of economic growth should also include the social dimension that reflects the construction of human capital, namely long-term healthy life and access to good education, which in our paper was represented by the regional medical level and regional college student ratio respectively.

Moreover, we further add another dimension, i.e., the environmental factors in our QGI to represent the sustainability of economic development. We use the emission of regional sulfur dioxide and industrial wastewater discharge to measure it.

Finally, we use the principal component analysis method to evaluate the QGI of each city to obtain the "overall" index value, which can reflect the regional sustainable economic growth. We employ the index data of 281 cities at the prefecture level from 2008 to 2017, and all the data are from the China Statistical Yearbook.

### 3.2.2. Explanatory Variable

We use the interaction dummy variable of whether the city invests in PPP projects $PPP_i * post_t$ as the main explanatory variable. We also use the annual investment amount of each city's PPP projects (*lninv_ppp*) as the second explanatory variable for testing. All the PPP project data comescome from the China Public Private Partnerships Center.

### 3.2.3. Control Variables

Aiming to make the estimation results more reliable, we select the urban characteristic variables that may affect the quality of regional economic growth, referring to Chen Shiyi and Chen Dengke [53] as control variables. The financial development (*lnfre*). The quality of regional economic growth is often closely related to the local financial development [30]. The regional financial development is measured by per capita loan balance of financial institutions. The government R&D investment (*lngyf*) measured by the proportion of fiscal

science and technology expenditure in total fiscal expenditure. The internet popularity (*lnhlw*). The internet is the basis of the development of digital economy and digital finance, and digital finance can effectively improve the mismatch of resources, narrow the income gap between urban and rural areas, and promote the improvement of the quality of economic growth; the internet popularity is the amount of internet access per capita. The industrial structure (*lnstr*) industrial structure optimization helps to promote high-quality development of economy [18]. It is measured by the ratio of employment in the tertiary industry and the secondary industry. The population density (*lnden*), the agglomeration effect brought about by cities with high population density, is more conducive to improving urban production efficiency and promoting the quality of economic growth [54]. The urban population density is measured by the population per unit area. All of these variables are logarithmically processed to eliminate the influence of heteroscedasticity, and the data are obtained from the China Statistical Yearbook. The descriptive statistics of various variables are shown in Table 2.

**Table 2.** Descriptive statistics of variables.

| Variables | Mean | SD | Min. | Max. | Sample Size |
|:---:|:---:|:---:|:---:|:---:|:---:|
| *QGI* | 4.942 | 0.087 | 3.839 | 5.191 | 2810 |
| *lninv_ppp* | 2.648 | 0.948 | −0.546 | 6.891 | 2810 |
| *lnrfe* | 0.641 | 0.404 | −0.601 | 2.175 | 2810 |
| *lngyf* | −0.448 | 1.051 | −5.030 | 2.582 | 2810 |
| *lnhlw* | −2.771 | 1.096 | −10.064 | 0.588 | 2810 |
| *lnstr* | 0.195 | 0.583 | −1.185 | 1.671 | 2810 |
| *lnden* | 5.896 | 0.973 | 1.664 | 8.864 | 2810 |

## 4. Results and Analysis

### 4.1. DID Estimated Results

Table 3 shows the DID estimation results. In Table 3, column (2) and column (5) add control variables on the basis of column (1) and column (4) models, respectively, and column (3) and column (6) build a robustness test by repeating 1000 times of random sampling on the basis of column (2) and column (5). All the models control both the time-fixed effect and the city-fixed effect. The estimated result shows the QGI in the cities with MCPs investment is 0.114 higher than that in the cities without MCPs investment after the PPP projects policies. The coefficient is significant at the level of 5%. The results are almost the same after the robustness test by repeating 1000 times random sampling. This finding shows that the investment of MCPs can significantly improve the quality of regional economic growth.

**Table 3.** The estimation results of DID strategy.

| Explained Variable: *QGI* | (1) | (2) | (3) | (4) | (5) | (6) |
|:---:|:---:|:---:|:---:|:---:|:---:|:---:|
| *PPP* × *post* | 0.116 *** | 0.114 ** | 0.114 ** | | | |
| | (0.06) | (0.06) | (0.05) | | | |
| *lninv_ppp* × *post* | | | | 0.026 *** | 0.025 *** | 0.025 *** |
| | | | | (0.01) | (0.01) | (0.01) |
| Control variables | N | Y | Y | N | Y | Y |
| City fixed effects | Y | Y | Y | Y | Y | Y |
| Year fixed effects | Y | Y | Y | Y | Y | Y |
| Bootstrap 1000 robustness standard error | N | N | Y | N | N | Y |
| N | 2810 | 2810 | 2810 | 2810 | 2810 | 2810 |
| R² | 0.264 | 0.297 | 0.297 | 0.290 | 0.320 | 0.320 |

Note: Standard errors in parentheses. ** $p < 0.05$, *** $p < 0.01$.

Different amounts of MCP investments may have different impacts on the quality of regional economic growth. In order to further explore the influence of the total investment amount of each city, we interact the investment amount variable *lninv_ppp* with the policy dummy variable *post* to substitute for the DID interaction item for estimation. The results show that there is a significant positive correlation between the investment amount of MCPs and the quality of regional economic growth. According to the estimated result, one unit of PPP project investment amount increase after the PPP policies will result in an increase of 0.025 in the regional economic growth, and the coefficient is significant at the level of 1%. Therefore, we can conclude that the MCPs investment can positively promote the regional economic growth quality.

### 4.2. Common Trend Test

The common trend hypothesis is an effective premise for DID estimation. We use the regression method to test the common trend. We expect that before the PPP projects investing policies, the different time trends are not related to the quality of economic growth across regions. We interact the time dummy variables with the policy dummy variable of the experimental and control groups for regression estimation. Table 4 reports the result. We can find that the coefficients of all cross-multiplication terms were not significant before 2014, while the coefficients of all cross-multiplication terms from 2014 to 2017 became significant, which validates the common trend hypothesis of our DID strategy.

**Table 4.** Common trend test.

| Explained Variable: *QGI* | Common Trend |
|---|---|
| *year 2008 × PPP* | 0.196 |
| | (0.137) |
| *year 2009 × PPP* | 0.184 |
| | (0.176) |
| *year 2010 × PPP* | 0.081 |
| | (0.150) |
| *year 2011 × PPP* | 0.123 |
| | (0.192) |
| *year 2012 × PPP* | 0.102 |
| | (0.181) |
| *year 2013 × PPP* | 0.168 |
| | (0.117) |
| *year 2014 × PPP* | 0.137 ** |
| | (0.066) |
| *year 2015 × PPP* | 0.139 ** |
| | (0.065) |
| *year 2016 × PPP* | 0.122 *** |
| | (0.056) |
| *year 2017 × PPP* | 0.137 ** |
| | (0.064) |
| Control variable | Y |
| City fixed effect | Y |
| Time fixed effect | Y |
| N | 2810 |
| $R^2$ | 0.483 |

Note: Standard errors in parentheses. ** $p < 0.05$, *** $p < 0.01$.

### 4.3. PSM–DID Estimation

Although the general DID method is used to compare the differences between the experimental group and the control group before and after the implementation of the policy, it still cannot solve the problem of systematic differences in the initial characteristics of cities before the implementation of the policy. Due to the significant differences in cities, governments may choose the cities with specific characteristics and avoid others when

investing in MCPs, which can also cause a bias in our estimation. The Propensity Score Matching method proposed by Rosenbaum and Rubin [55] can be used to overcome the possible sample selection bias. The PSM–DID model specification is as follows:

$$QGI_{it}^{PSM} = \alpha_0 + \alpha_1 PPP_i * post_t + \alpha_2 X_{it} + \delta_t + \mu_i + \varepsilon_{it} \tag{2}$$

The equilibrium hypothesis of the matched samples needs to be tested before using the PSM–DID method for estimation. The matching effect is tested by the standard deviation of the samples: if the absolute value of the standard deviation after matching is less than 20%, the matching process is effective. Table 5 presents the result of our test of matching, which shows the standard deviation of the main variables before and after matching. The standard deviation of all variables after matching is below 16%, which proves the effectiveness of the matching process in this paper.

**Table 5.** Test of matching results of variable propensity score.

| Variable | Match or Not | Mean | | Error (%) |
| | | Experimental Group | Control Group | |
|---|---|---|---|---|
| *lnrfe* | U | 0.720 | 0.623 | 21.5 |
| | M | 0.697 | 0.649 | 10.7 |
| *lngyf* | U | −0.311 | −0.770 | 39.3 |
| | M | −0.365 | −0.356 | −0.8 |
| *lninf* | U | 3.399 | 3.777 | −53.3 |
| | M | 3.431 | 3.427 | 0.6 |
| *lnhlw* | U | −2.760 | −3.308 | 44.3 |
| | M | −2.836 | −2.949 | 9.1 |
| *lnstr* | U | 0.121 | 0.361 | −35.2 |
| | M | 0.139 | 0.244 | −15.5 |
| *lnden* | U | 6.607 | 5.746 | 80.9 |
| | M | 6.555 | 6.607 | −4.9 |

We then conduct the PSM–DID estimation, and the results are reported in Table 6. We can find that the coefficient of the DID interaction item is 0.114 and significant at the 5% level, which stands almost the same as the basic DID results. We also test the amount of MCPs investments, and the result is also the same as that of DID strategy. Therefore, the results of the PSM–DID estimation confirm the conclusion that the MCPs investment can improve the quality of regional economic development.

**Table 6.** The estimation results of the PSM–DID strategy.

| Explained Variable: *QGI* | (1) | (2) | (3) | (4) |
|---|---|---|---|---|
| *PPP* × *post* | 0.115 ** | 0.114 ** | | |
| | (0.06) | (0.06) | | |
| *lninv_ppp* × *post* | | | 0.025 *** | 0.025 *** |
| | | | (0.01) | (0.01) |
| Control variables | N | Y | N | Y |
| City fixed effects | Y | Y | Y | Y |
| Year fixed effects | Y | Y | Y | Y |
| N | 2810 | 2810 | 2810 | 2810 |
| R$^2$ | 0.268 | 0.277 | 0.320 | 0.320 |

Standard errors in parentheses. ** $p < 0.05$, *** $p < 0.01$.

### 4.4. Robustness Test

We further conduct a series of robustness to verify the reliability of our conclusion. Table 7 reports the results of the robustness test.

**Table 7.** Robustness test analysis.

| Explained Variable: *QGI* | (1) | (2) | (3) | (4) | (5) | (6) | (7) | (8) |
|---|---|---|---|---|---|---|---|---|
| *PPP* × *post* | 0.110 *** | 0.111 *** | 0.114 *** | 0.116 *** | 0.110 *** | 0.094 *** | 0.111 *** | 0.114 *** |
| | (0.05) | (0.05) | (0.06) | (0.03) | (0.02) | (0.02) | (0.05) | (0.04) |
| Control variables | Y | Y | Y | Y | Y | Y | Y | Y |
| City fixed effects | Y | Y | Y | Y | Y | Y | Y | Y |
| Year fixed effects | Y | Y | Y | Y | Y | Y | Y | Y |
| N | 2810 | 2810 | 2810 | 2810 | 2660 | 2600 | 2580 | 2710 |
| $R^2$ | 0.254 | 0.257 | 0.257 | 0.280 | 0.350 | 0.350 | 0.359 | 0.312 |

Standard errors in parentheses. *** $p < 0.01$.

Since the quality of regional economic growth may still be affected by some non-observable factors that vary with time and region, the cross-multiplication term of province dummy and time trend and the interaction effect of province dummy and tripled time trend are added into columns (1) and (2), respectively. In addition, the intersection term of province fixed effect and year fixed effect is controlled in column (3).

In column (4), we further cluster the estimated standard errors at the provincial level. Considering that there are obvious differences in economic development and government supervision among different city levels, we classify cities by excluding the sub-provincial cities according to the administrative hierarchy stipulated. The results are shown in column (5).

In column (6), the outlier values of QGI samples are removed. Specifically, according to the percentile distribution of QGI in the whole sample, the extreme value samples at the 1% and 99% percent are eliminated; that is, the city samples with QGI in the highest score group and the lowest quantile group are excluded from the test. Column (7) excludes the outlier of the PPP project investment amount, and column (8) deletes the outlier of the population size using the same method as in column (6).

From Table 7, we can find that all the estimated coefficients are positively significant at the level of 1%, which proves the robustness of the conclusion of our study.

## 5. Mechanism Test

Although we have verified the positive effects of MCPs investments on regional economic growth quality, we still wish to figure out in what way MCP investments specifically promote the QGI, which will help us to understand the relationship between the MCPs investment and regional economic growth quality much better.

### 5.1. Strategy and Model

According to the literature review section, we tend to test the mechanism from the regional innovation entrepreneurship aspect. We take the number of regional patent applications (*lnEnte*, logarithm form) as the proxy variable of regional entrepreneurial innovation referred to by Hongbin Li [56], etc. Based on the model (1) and model (2), we construct a triple interaction term of the previous DID item and the *lnEnte* variable for estimation, and the model is as follows:

$$QGI_{it}^{PSM} = \alpha_0 + \alpha_1 PPP_i * post_t * lnEnte + \alpha_2 X_{it} + \delta_t + \mu_i + \varepsilon_{it} \tag{3}$$

### 5.2. Result Analysis

Table 8 reports the mechanism test results. Column (1) is the estimated result of DID strategy, and column (2) is the estimated result of PSM–DID strategy. All the coefficients of the multiplicative term are positively significant at the 1% level, which shows that the investment of MCPs can highly improve the regional entrepreneurial innovations and hence promote the economic growth quality. A good entrepreneurial shaping environment can be created through the investments of MCPs, and higher regional entrepreneurial innovations may promote a higher quality of economic development. Based on the results shown in

Table 8, we confirm the possible mechanism that the MCPs investments can promote the regional economic growth quality by enhancing the regional innovation entrepreneurship.

**Table 8.** The results of mechanism test.

| Explained Variable: *QGI* | (1) DID | (2) PSM–DID |
|---|---|---|
| $PPP \times post \times lnEnte$ | 0.041 *** | 0.047 *** |
| | (0.01) | (0.01) |
| $PPP \times post$ | 0.052 | 0.055 |
| | (0.06) | (0.06) |
| *lnEnte* | 0.114 ** | 0.126 ** |
| | (0.07) | (0.07) |
| Control variables | Y | Y |
| City fixed effects | Y | Y |
| Year fixed effects | Y | Y |
| N | 2810 | 2810 |
| $R^2$ | 0.295 | 0.297 |

Standard errors in parentheses. ** $p < 0.05$, *** $p < 0.01$.

## 6. Conclusions and Implication

In order to investigate the overall effect of the MCPs investment on regional economic growth, this paper constructs a comprehensive regional economic growth quality index including sustainable development factors based on the conclusion of Mlachila et al. [26] and takes the PPP projects policies since 2014 in China as a quasi-natural experiment to design a DID strategy to identify the exogenous impacts of MCPs investments on regional economic growth for the first time. The estimation results show that the MCPs investment will significantly improve the overall quality of regional economic growth and also be conducive to the long-run sustainable development. After a series of robustness tests, including the common trend test and the PSM–DID test, our conclusions still hold. Moreover, we also reveal the mechanism that the MCPs investments will promote the quality of regional economic growth by enhancing the regional innovation entrepreneurship.

Different from existing studies which mainly focused on the short-term influence on the quantitative GDP increase or the long-term impact on some specific aspects of economic growth, we figure out "overall" effects of the MCPs investment on sustainable economic development, which is helpful for governments to make the investment policy in the infrastructure constructions to promote the regional sustainable development, especially in current economic recession after the COVID-19 pandemic. In addition, our DID strategy of taking the PPP policy as a quasi-natural experiment for the identification of MCP investment is conducive to further research on the effects of the MCPs on other social or environmental aspects.

Our findings have the following policy implications. Under the influence of the current downward pressure on the economy, the government can encourage private capital to enter the public construction fields by flexible use of monetary policy, fiscal policy, and other policies. Thus, the government can fully play the counter-economic cyclical adjustment role by investing in MCPs. Furthermore, the government can incorporate the commissioning of MCPs with some emerging technologies in order to introduce more innovation resources into the MCPs and help to enhance more regional innovation entrepreneurship for a higher quality of economic growth.

Although we empirically confirm the conclusion that MCPs investment can promote the quality of regional economic growth, the different investment modes or different types of MCPs may also change those effects, which is not involved in this paper. Future research can further explore the impacts of the various modes and types of MCPs on economic growth.

**Author Contributions:** Conceptualization, F.W. and H.Y.; Data curation, X.H. and H.G.; Formal analysis, H.Y.; Funding acquisition, M.Y.; Methodology, F.W., X.H. and P.Z.; Visualization, M.Y.; Writing—original draft, P.Z. and H.Y. All authors have read and agreed to the published version of the manuscript.

**Funding:** This research received no external funding.

**Institutional Review Board Statement:** Not applicable.

**Informed Consent Statement:** Informed consent was obtained from all subjects involved in the study.

**Data Availability Statement:** The data used to support the findings of this study are available from the corresponding author upon request.

**Conflicts of Interest:** The authors declare no conflict of interest.

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
