# Peer review of "The Effects of Investment in Major Construction Projects on Regional Economic Growth Quality: A Difference-In-Differences Analysis Based on PPP Policy"

_sustainability, doi:10.3390/su14116796_

Round 1

Reviewer 1 Report

In the abstract: *major construction projects(MCPs), please change to Major                                 Construction Projects(MCPs)

                         *quality growth index (QGI), please change to Quality                                         Growth Index (QGI)

                          *difference-in-differences(DID), please change to Difference-In-Differences(DID)

The writing of the paper needs a lot of improvement in terms of grammar, spelling, and presentations. The paper needs careful English polishing since there are many typos and poorly written sentences.

Some examples are as the following:

*     Check the usage of the commas carefully.

*     Check the articles including "a", "an" and "the".

*     Check the required and unneeded blank spaces.

 The part of the abstract is not attractive

The introduction you can present better novelty of your work

Literature analysis can be improved.

Avoid repetitions. I can see several repetitions at different places in this paper. A thorough proofreading is required.

Contributions are unclear. This should be made clear from the very beginning of the paper till its end.

There is no scope for future research. A clear direction for future research is required.

Author Response

Dear reviewer:

Thank you so much for your comments! Those comments are very helpful in improving our paper. According to your comments, we have substantially revised our manuscript. Now our responses to each comment are in the attached word file.

Thank your for your comments once again.

Best regard

Reviewer 2 Report

The manuscript investigated the effect of major construction project on economy growth. The manuscript is well written and organized. Since the Journal is Sustainability, the authors need to address how the manuscript is related to sustainability which is missing in the current version. 

Author Response

Dear reviewer:

Thank you so much for your comments! Those comments are very helpful in improving our paper. According to your comments, we have substantially revised our manuscript. Now our responses to each comment are in the attached word file.

Thank you once again.

Best regards

Reviewer 3 Report

Introduction

  1. The investment of MCPs serves as a policy tool ~ : this sentence can be revised in a direct form. The current sentence is a little awkward
  2. In the sentence of “Many scholars ~”, “from different perspective of departments: the expression is not scientific and clear.
  3. “The investment of MCPs has ~ promoted the …” > what kinds of regional economy / how can it has promoted?
  4. “Nevertheless, ..” Do you have any evidence (citations)?
  5. Is it necessary to ~ > why is it necessary? The previous content did not support the reason of the necessity.
  6. Since the necessity is not persuasive, the rest of introduction should be revised.

Policy background and theoretic mechanism

  1. What tare he PPP projects drastically improved in 2015?
  2. Some of expressions (believed, etc.) are not scientific.
  3. “Consistent with ~, we provide ~.” > then, whare is a significance of this research?

Methodology and Data

  1. The explanation of variables and model are not sufficient for providing the methodology. The reason the authors adapted the certain methodology should be clearly described.
  2. The section 3.2 is not read as methodology, and it is just the beginning of analysis (about testing) not methodology

Analysis

  1. The authors divided analysis into methodology and analysis sections. The readers should be able to clearly understand the research methodology in the methodology section. Then when professionals adapt the same / similar methodology provided in this research, they should be able to get similar results in analysis. In terms of this notion, the description in the manuscript does not seems to logical / scientific.

Mechanism test and heterogeneity analysis

  1. What are the reasons of mechanism test and heterogeneity analysis? What are the purpose?

Conclusions and Enlightenment

  1. Are there any counter-intuitive results / conclusions this research is discovered?
  2. Please precisely describe the limitations and future research. The suggestions of 5G infrastructure, etc. are not supported from a logical thought process.

Author Response

(The authors gave the same response as above.)

Round 2

Reviewer 1 Report

Accepted

Reviewer 3 Report

Thank you for addressing previous concerns.